# Vision LLMs Are Bad at Hierarchical Visual Understanding, and LLMs Are the Bottleneck

## Abstract

This paper reveals that many state-of-the-art large language models (LLMs) lack hierarchical knowledge about our visual world, unaware of even well-established biology taxonomies. This shortcoming makes LLMs a bottleneck for vision LLMs' hierarchical visual understanding (e.g., recognizing `Anemone Fish` but not `Vertebrate`). We arrive at these findings using about one million four-choice visual question answering (VQA) tasks constructed from six taxonomies and four image datasets. Interestingly, finetuning a vision LLM using our VQA tasks reaffirms LLMs' bottleneck effect to some extent because the VQA tasks improve the LLM's hierarchical consistency more than the vision LLM's. We conjecture that one cannot make vision LLMs understand visual concepts fully hierarchical until LLMs possess corresponding taxonomy knowledge.

## 1 Introduction

Taxonomy is natural and core in visual understanding. The biology taxonomies cover many objects in our visual world [53]; for example, a `Boston Terrier` belongs to the class of `Terrier`, which is a subtype of `Dog`, under `Mammal`, and ultimately part of the broader category `Animal`, forming a semantic path in the animal taxonomy: `Animal → Mammal → Dog → Terrier → Boston Terrier`. ImageNet [13] expands from the WordNet [37] taxonomy. Visual parts [28, 15, 3], attributes [14, 27, 41], and relationships [26] can be grouped hierarchically due to shared characteristics.

A high-performing, general-purpose visual understanding system should map visual inputs to both fine-grained leaf nodes of a taxonomy and coarse-grained inner nodes. Meanwhile, it should label an input hierarchically consistently along the path that traces a leaf up to the root. Figure 1 illustrates a case selected from our experiments that the model predictions lack *hierarchical consistency*, failing to follow the path of `Animal → Vertebrate → Fish → Spiny-finned Fish → Anemone Fish`.

Surprisingly, little has been done to assess the hierarchical visual understanding performance of vision large language models (VLLMs) [4, 29, 9, 72, 34, 29], which have the potential to make such a general-purpose vision system. Indeed, VLLMs unify various vision tasks (e.g., visual recognition [13], captioning [8], question answering [2], and retrieval [62]) into one model by anchoring visual encoders [46, 66, 10, 39] to a versatile pretrained LLM [19, 60], typically orders of magnitude bigger, offering integrated interactions with humans that involve images and videos in conjunction with natural language prompts. Comprehensively benchmarking VLLMs is essential for realizing their potential and identifying opportunities for improvements. Extensive benchmarks have recently emerged, such as the bilingual MMBench [36], manually labeled MME [16], and MMMU [64] collected from college exams. We refer readers to [57] for an extensive list.

This work systematically evaluates VLLMs' hierarchical visual understanding capabilities using six taxonomies and four hierarchical image classification datasets. Conventionally, the hierarchical im-

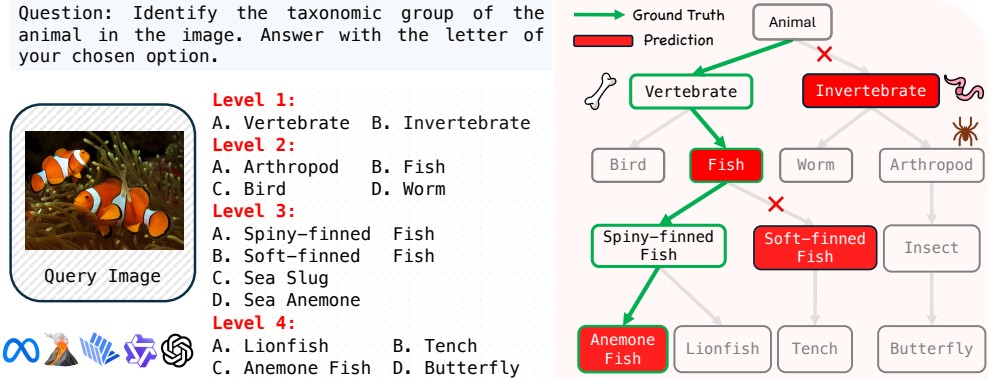

Figure 1: **Left**: Four-choice VQA tasks for evaluating VLLMs' hierarchical visual understanding. **Right**: A VLLM's answers ( in red boxes ) deviate from the ground truth path (**green arrows**), illustrating its lack of hierarchical consistency.

age classification [47, 44, 58, 61, 59] aims to classify visual inputs into semantically structured categories across multiple levels of specificity, in contrast to flat classification, which treats labels as mutually exclusive and unstructured. We construct about one million four-choice visual question-answering (VQA) tasks from the hierarchical datasets (see Figure 1 for some examples). The tasks traverse all taxonomy levels, and the four choices of an individual task are from the same level. When evaluating VLLMs' performance over these tasks, we stress hierarchical consistency because it is unique to hierarchical visual understanding and crucial for adaptability to users' varying granularity preferences [44, 12, 58].

Our main findings are as follows. First of all, many state-of-the-art VLLMs struggle with our VQA tasks, substantially lacking hierarchical consistency. For example, Qwen2.5-VL-72B [4] makes mistakes over 67% of the hierarchical paths in the iNaturalist [53] taxonomy. Moreover, in our attempt to tracing down the error causes, we find that LLMs are the bottleneck and lack taxonomy knowledge about the visual world. In contrast, the visual encoder and projector modules demonstrate the ability to retain highly discriminative and well-structured visual features. We further show that the LLM embeddings about the visual concepts contain sufficient hierarchical cues and organize them orthogonally, but the model cannot decode them. Finally, finetuning a VLLM using our VQA tasks enhance its LLM's (text) hierarchical consistency more than the VLLM's (visual) hierarchical consistency, reaffirming LLMs' bottleneck effect to some extent.

## 2 VLLMs Lack Hierarchical Consistency in Visual Understanding

We construct six hierarchical image classification benchmarks in a four-choice VQA format to systematically assess VLLMs' accuracy and hierarchical consistency in visual understanding. These benchmarks leverage datasets that inherently exhibit taxonomic structures, either derived from Word-Net [37] or grounded in biological classification standards. In what follows, we formally define hierarchical image classification, followed by two evaluation metrics about accuracy and consistency, respectively. We then describe our VQA tasks and the first set of experiment results in this work.

### 2.1 Hierarchical Image Classification: Notations and Problem Statement

General image classification tasks typically assume a flat label space, where each image $x \in \mathcal{X}$ is assigned a class label $y \in \mathcal{Y}$ out of a predefined set $\mathcal{Y}$ of mutually exclusive categories. However, many real-world problems exhibit rich semantic structures, in which labels are naturally organized into a hierarchy $\mathcal{T} = (\mathcal{Y}, \mathcal{E})$ [44, 58, 61, 59], such as a tree or a directed acyclic graph. Here, $\mathcal{E} \subseteq \mathcal{Y} \times \mathcal{Y}$ denotes the set of directed edges representing parent-child relationships, where $(y_i, y_j) \in \mathcal{E}$ indicates that $y_i$ is the parent of $y_j$ in the hierarchy. In hierarchical image classification, the objective is not only to predict the leaf node label $y \in \mathcal{Y}_{leaf} \subseteq \mathcal{Y}$ but also to correctly recover its full ancestral path $(y_0, y_1, \cdots, y_L)$ in $\mathcal{T}$, where $y_0$ denotes the root node and $L$ is the depth of the hierarchy. In this paper, we aim to evaluate VLLMs' hierarchical image classification capabilities, identify their limitations and underlying causes, and enhance their performance based on these insights.

Table 1: Overview of the six taxonomies and four datasets we use to construct the VQA tasks.

| Dataset | # Levels | # Leaf Nodes | # Test Images | Hierarchy Distribution |
|---|---|---|---|---|
| CUB-200-2011 [54] | 4 | 200 | 5,794 | 13-37-124-200 |
| iNaturalist-Plant [53] | 6 | 4,271 | 42.71K | 5-14-85-286-1702-4271 |
| iNaturalist-Animal [53] | 6 | 5,388 | 53.88K | 6-27-152-715-2988-5388 |
| ImageNet-Animal [13] | 11 | 397 | 19.85K | 2-10-37-81-123-81-65-41-64-34-2 |
| ImageNet-Artifact [13] | 8 | 492 | 24.60K | 5-40-149-205-162-62-44 |
| Food-101 [5] | 4 | 84 | 21.00K | 6-29-40-24 |

## 2.2 Two Evaluation Metrics about Accuracy and Consistency, Respectively

For evaluation, we mainly focus on the hierarchical consistency of model predictions [58, 43]. Besides, we are interested in the leaf-level classification accuracy [68, 35, 20], which can be viewed as the upper bound of the hierarchical consistency, detailed below.

**Hierarchical Consistent Accuracy (HCA) [58, 43].** This metric is defined as

$$\text{HCA} = \frac{1}{N} \sum_{i=1}^{N} \prod_{j=1}^{L^i} \mathbb{1} \left[ f_\theta \left( x^i ; \mathcal{Y}_j \right) = y_j^i \right],$$ (1)

where $N$ is the number of images in the testing set, $L^i$ denotes the depth of the hierarchy for the $i$-th input $x^i$ and may vary for different tasks in uneven trees, $f_\theta : \mathcal{X} \mapsto \mathcal{Y}$ is an image classifier, $\mathcal{Y}_j$ represents the set of labels at the $j$-th layer of the hierarchy, and $\mathbb{1}[\cdot]$ is an indicator function. HCA evaluates whether a model's predictions are consistent with the entire hierarchical path from the root to a leaf node. Specifically, it measures the proportion of samples for which all ancestor nodes along the predicted paths match the ground truth. This is a stricter metric than flat accuracy and serves as our primary evaluation criterion for hierarchical classification.

**Leaf-Level Accuracy** $\text{Acc}_{\text{leaf}}$ **[68, 35, 20].** It cares about the predictions at the most fine-grained level of a taxonomy:

$$\text{Acc}_{\text{leaf}} = \frac{1}{N} \sum_{i=1}^{N} \mathbb{1} \left[ f_\theta \left( x^i ; \mathcal{Y}_L \right) = y_L^i \right].$$ (2)

Interestingly, $\text{Acc}_{\text{leaf}}$ upper-bounds HCA because correctly assigning a leaf label $y_L$ to an input $x$ contributes to $\text{Acc}_{\text{leaf}}$, but it does not increase HCA unless the model makes no mistake over all nodes in the path $(y_0, y_1, \cdots, y_L)$ connecting the leaf label to the root.

## 2.3 VQA Tasks Derived from Hierarchical Image Classification Datasets

VLLMs are the image classifiers $f_\theta$ in equations (1) and (2), and one can use language prompts to steer their output to a particular taxonomy level. More concretely, we formalize a VQA task for each image given a desired taxonomy level, $(x^i, \mathcal{Y}_j), i = 1, 2, \cdots, N, j = 1, 2, \cdots, L^i$, as follows.

**VQA Tasks.** We derive approximately one million four-choice VQA tasks and six taxonomies from four hierarchical image classification datasets [54, 53, 13, 5] to evaluate VLLMs in a closed-set setting. This setting mitigates the challenge of open-set generation, which involves a prohibitively large prediction space [68] and ambiguous prediction granularity. We test different VQA prompts (provided in Appendix C), and they generally follow this format:

```
<image> Given the plant in the image, what is its taxonomic classification
at the <hierarchy> (e.g., kingdom) level?
A.<similar class> B.<ground truth> C.<similar class> D.<similar class>
Answer with the option letter only.   (Choices are shuffled in the experiments)
```

Arguably, the four-choice VQA tasks are easier than the conventional hierarchical image classification, whose label space is orders of magnitude bigger than four. To compensate this difference, we make sure the four choices are from the same level of a taxonomy and use "confusing labels" in the VQA tasks. Specifically, we use SigLIP [66] to compute the cosine similarity scores between an image and all text labels other than the ground truth (at a particular taxonomy level), selecting

Table 2: The hierarchical consistent accuracy (HCA) and leaf-level accuracy $\text{Acc}_{\text{leaf}}$ of six open-source VLLMs, two CLIP-style models, and the proprietary GPT-4o.

| Model | iNat21-Animal | | iNat21-Plant | | ImgNet-Artifact | | ImgNet-Animal | | CUB-200 | |
|---|---|---|---|---|---|---|---|---|---|---|
| | HCA | $\text{Acc}_{\text{leaf}}$ | HCA | $\text{Acc}_{\text{leaf}}$ | HCA | $\text{Acc}_{\text{leaf}}$ | HCA | $\text{Acc}_{\text{leaf}}$ | HCA | $\text{Acc}_{\text{leaf}}$ |
| **Open-Source VLLMs** | | | | | | | | | | |
| LLaVA-OV-7B [29] | 4.53 | 26.47 | 4.46 | 27.51 | 17.15 | 80.77 | 34.36 | 65.50 | 11.51 | 44.23 |
| InternVL2.5-8B [9] | 8.52 | 27.65 | 5.56 | 28.36 | 21.42 | 78.07 | 37.82 | 65.19 | 22.07 | 45.56 |
| InternVL3-8B [72] | 11.93 | 35.40 | 8.68 | 36.39 | 17.87 | 77.50 | 42.31 | 69.41 | 25.75 | 50.52 |
| Qwen2.5-VL-7B [4] | 19.43 | 41.33 | 17.67 | 41.61 | 16.47 | 85.20 | 56.00 | 80.01 | 43.76 | 65.50 |
| Qwen2.5-VL-32B [4] | 26.90 | 46.98 | 24.64 | 48.57 | 26.30 | 84.51 | 62.23 | 80.48 | 56.80 | 69.00 |
| Qwen2.5-VL-72B [4] | 35.73 | 54.20 | 32.82 | 55.00 | 21.08 | 85.61 | 64.08 | 80.52 | 66.36 | 75.04 |
| **CLIP Models** | | | | | | | | | | |
| OpenCLIP [10] | 1.04 | 23.53 | 0.19 | 28.12 | 9.11 | 83.64 | 12.57 | 81.14 | 4.31 | 80.39 |
| SigLIP [66] | 2.15 | 12.71 | 0.46 | 18.84 | 6.41 | 87.19 | 24.40 | 86.85 | 23.18 | 73.84 |
| **Proprietary VLLM** | | | | | | | | | | |
| GPT-4o [1] | 42.95 | 63.79 | 35.53 | 62.95 | 27.57 | 86.05 | 67.69 | 85.50 | 81.96 | 87.25 |

the top three most similar labels as the distracting VQA choices. Besides, we provide the results of randomly sampled choices in Appendix B.

**Hierarchical Image Classification Datasets.** Table 1 summarizes the six taxonomies and four datasets we use to construct the VQA tasks. CUB-200-2011 (CUB-200) [54] is a fine-grained bird dataset containing 200 species. We prompt GPT-4o [1] to map each class to a four-level taxonomy: Order → Family → Genus → Specie. To ensure taxonomic accuracy, we cross-validate the generated hierarchy using corresponding entries from Wikipedia. In addition, we incorporate the iNaturalist-2021 (iNat21) dataset [53], a large-scale collection with species-level annotations spanning various biological taxa. We separate it into two taxonomies, Plant and Animal, comprising 4,271 and 5,388 leaf nodes, respectively, and six levels. Both CUB-200 and iNat21 provide well-established biological taxonomies with even hierarchical depths. To increase structural diversity, we also experiment with ImageNet-1K (ImgNet) [13], whose leaf labels are coarser-grained than iNat21 and CUB-200. ImgNet is built upon the WordNet [37]. We extract two relatively well-structured subsets from ImgNet: ImgNet-Animal and ImgNet-Artifact, following [58]. We further refine these subsets to improve label quality and semantic consistency. Food-101 [5] is about food classification, and its hierarchy is constructed based on the recent work of Liang and Davis [32].

## 2.4 Experiments and Findings

We mainly study state-of-the-art open-source VLLMs: The Qwen2.5-VL [4] models of 7B, 32B, and 72B parameters, InternVL2.5-8B [9], InternVL3-8B [72], and LLaVA-OV-7B [29]. Meanwhile, we include the proprietary GPT-4o's results for reference; in general, GPT-4o slightly outperforms Qwen-2.5-VL-72B, but the main findings below still apply. Finally, we experiment with two CLIP-style [46] models, SigLIP-SO400M [66] and OpenCLIP-L [10], following the experiment protocol in [46] except that the candidate labels for each test image are restricted to the same four choices as fed to VLLMs. Table 2 shows the results about the models' hierarchical consistency (HCA) and leaf-level accuracy ($\text{Acc}_{\text{leaf}}$) on iNat21, ImgNet, and CUB-200. The Food-101 results are in Appendix B to save space in the main text. We draw the following conclusions.

**VLLMs Lack Hierarchical Consistency in Visual Understanding.** Regardless of the leaf-level accuracy, all open-source VLLMs, CLIP models, and GPT-4o lack hierarchical consistency because their HCA is significantly lower than $\text{Acc}_{\text{leaf}}$ (up to 99.3% relatively). The gaps on iNat21-Plant are especially big (e.g., 32.82 vs. 55.00 for Qwen2.5-VL-72B and 35.53 vs. 62.95 for GPT-4o). While one might expect better results on ImgNet, neither open-source VLLMs nor GPT-4o can make their HCA match $\text{Acc}_{\text{leaf}}$ — more than 20% decrease for all models, indicating that VLLMs make many mistakes along the paths from the taxonomies' roots to the leaf nodes even when they are correct over the leaves.

**Fine-Grained Visual Recognition Remains Challenging for VLLMs.** While VLLMs and CLIP models perform moderately well on ImgNet, they struggle with fine-grained object recognition; on

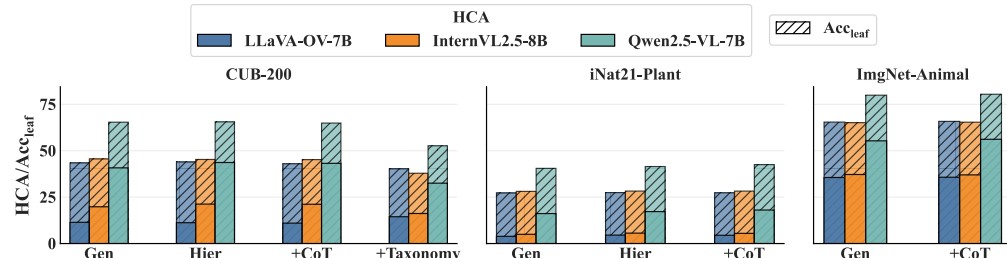

Figure 2: Prompt variants and their effects on VLLMs' hierarchical consistency (HCA) and fine-grained recognition $\text{Acc}_{\text{leaf}}$ (**Gen**: general prompts, **Hier**: hierarchical prompts, **+CoT**: prompts with Chain-of-Thought reasoning, **+Taxonomy**: prompts that include an explicit taxonomy in the JSON format. Please see Appendix C for details and examples.).

the iNat21 dataset, even the best-performing GPT-4o gives rise to only 63% leaf-level accuracy, far from its 86% on ImgNet. Notably, InternVL2.5 and LLaVA-OV's results (about 27%) on iNat21 are only slightly above random guess (25%), and the CLIP models are barely on par with random guess. In contrast, a small task-specialized model [23] leads to 61.56% leaf-level accuracy on iNat21, and some models [11, 69] achieve 93% accuracy on CUB-200, outperforming all the general-purpose VLLMs in our experiments. These findings are consistent with the recent work [17, 68, 20, 63] that recognizes the limitation of VLLMs on (fine-grained) image classification.

**Scaling Laws Works for Hierarchical Visual Understanding.** Both hierarchical consistency and leaf-level accuracy improve as the size of the Qwen2.5-VL series of models increases. Moreover, the gap between HCA and $\text{Acc}_{\text{leaf}}$ progressively narrows. However, the largest models (Qwen2.5-VL-72B and GPT-4o) are still unsatisfactory in terms of both hierarchical consistency and fine-grained recognition, especially on the iNat21 benchmark.

**Qwen2.5-VLs Are Among the Most Powerful Open-Source VLLMs.** LLaVA-OV-7B's hierarchical consistency and leaf-level accuracy are below InternVLs and Qwen2.5-VLs. InternVL3-8B improves upon InternVL2.5-8B, but it is still under par with Qwen2.5-VL-7B.

## 3 Why Are VLLMs Poor at Hierarchical Image Classification?

We systematically investigate potential causes of VLLMs' low performance on hierarchical visual understanding. We first extensively study prompt variations in Section 3.1 and reveal that some prompts can lead to marginally better results than the rest, but the results remain generally bad. We then examine VLLMs' visual encoders and subsequent visual tokens to see whether and where essential visual information is lost when it forwards through VLLMs (Section 3.2). Interestingly, the discriminative cues in the visual tokens are maintained across various stages of the VLLM architectures, leading to about the same hierarchical image classification results immediately after the visual encoder, after the projection to the language token space, and at the very last layer of an LLM. Finally and surprisingly, we find that the generally believed powerful LLMs, even the one with 72B parameters in our experiments, lack basic taxonomy knowledge and are likely responsible for VLLMs' poor performance on hierarchical visual understanding! (We believe this conclusion is true for open-source VLLMs, but we urge readers not to extrapolate it to proprietary LLMs because we could not probe their intermediate embeddings.)

### 3.1 Language Prompts Are *Not* the Bottleneck

Prompt engineering often comes as a remedy for boosting VLLMs' performance in different applications [6, 55, 68, 58]. Could it also rescue VLLMs on our hierarchical visual understanding tasks? We strive to test prompt variants comprehensively. We specify the taxonomy levels in the prompts for CUB-200 [54] and iNat21 [53], whose taxonomies are grounded in biology. We even add CUB-200's complete taxonomy as a JSON file to the prompts. For the other datasets with more generic taxonomies, we test general and chain-of-thought [24, 57] prompts derived from the template in Section 2.3. Appendix C provides all prompts in detail, and Figure 2 shows the results of some high-performing prompts. We can see from the results that the prompt design alone is insufficient to improve VLLMs' hierarchical consistency or leaf-level accuracy.

Table 3: (Text) HCA of VLLMs' LLMs and its correlation $\rho$ with VLLMs' (visual) HCA

| LLM of | iNat21-Animal | iNat21-Plant | ImgNet-Artifact | ImgNet-Animal | CUB-200 | $\rho$(text,visual) |
|---|---|---|---|---|---|---|
| LLaVA-OV-7B [29] | 11.56 | 28.49 | 29.27 | 56.93 | 33.45 | 0.9116 |
| InternVL2.5-8B [9] | 38.15 | 41.15 | 35.32 | 66.11 | 49.11 | 0.8832 |
| InternVL3-8B [72] | 54.20 | 47.49 | 31.86 | 69.92 | 59.87 | 0.9030 |
| Qwen2.5-VL-7B [4] | 52.08 | 64.21 | 35.06 | 68.14 | 63.86 | 0.8640 |
| GPT-4o [1] | 96.85 | 96.70 | 42.31 | 89.56 | 98.81 | 0.7980 |

## 3.2 Visual Embeddings Are *Not* the Bottleneck

The open-source VLLMs in this work vary in specific implementations, but their core components are the same: A visual encoder mapping images to embeddings, a projector translating visual embeddings into the language token space, and an LLM. If the hierarchical structure and discriminativeness are lost before the visual embeddings reach LLMs, the overall VLLMs would inevitably perform poorly on our hierarchical visual understanding tasks. Hence, it is crucial to examine the visual embeddings. We train three linear classifiers per taxonomy level to respectively probe the visual encoder, projector, and last layer of an LLM, where the image representations are an average of the visual tokens. Further details and results of the probing are provided in Appendix C.

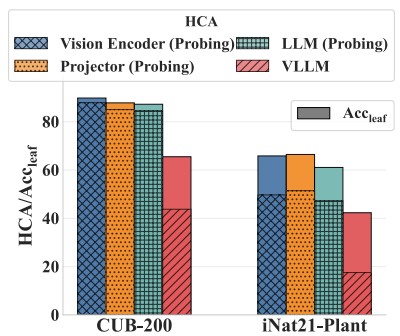

Figure 3: Qwen2.5-VL-7B vs. linearly probing the visual tokens at various stages of Qwen2.5-VL-7B on CUB-200 and iNat21-Plant.

Figure 3 shows the probing results of Qwen2.5-VL-7B over CUB-200 [54] and iNat21-Plant [53]. Remarkably, the linear classifiers outperform Qwen2.5-VL-7B all around. They achieve not only higher leaf-level accuracy than Qwen2.5-VL but also much better hierarchical consistency, even though the classifiers of different taxonomy levels are independently trained. Moreover, the linear probing results remain about the same at different stages of the forward propagation (i.e., immediately after the visual encoder, projector, and last layer of the VLLM), indicating that the visual tokens remain discriminative and structurally rich throughout different LLM layers. These results are a strong defense for the visual embeddings: They carry sufficient hierarchical and discriminative cues and should not be blamed for VLLMs' poor hierarchical visual understanding performance.

## 3.3 *LLMs Are the Bottleneck* in VLLMs' Hierarchical Visual Understanding

The huge discrepancy between the results of linearly probing visual tokens and VLLM performance in Figure 3 propels us to investigate other potential causes of VLLMs' low hierarchical consistency beyond the visual embeddings, and we find that the influential LLMs are the bottleneck.

### 3.3.1 Open-Souce VLLMs' LLMs Lack Taxonomy Knowledge

We separate LLMs from open-source VLLMs and examine how much they know about the taxonomies used in our experiments. Mechanically, we reformulate our VQA tasks to a text-only version by replacing the images with their corresponding leaf labels:

```
Given the <leaf node label> (e.g., Anemone Fish), what is its taxonomic
classification at the <hierarchy> (e.g., kingdom) level?
A.<similar class>  B.<ground truth>  C.<similar class>  D.<similar class>
Answer with the option letter only.  (Choices are shuffled in the experiments)
```

This process results in about 0.7 million QA tasks after deduplication. We use them to assess LLMs and report the (text) HCA results in Table 3 — we use (text/visual) HCA to refer to LLMs/VLLMs' performance on text/visual QA tasks for clarity. We find that Qwen2.5-VL-7B's LLM achieves only 63.86% (text) HCA on CUB-200, whose taxonomy comprises merely four levels. The LLMs of LLaVA-OV and InternVL-2.5 give rise to even lower (text) HCAs on CUB-200 (33% and 49%). One might wonder if these low (text) HCAs are due to that the biology taxonomy underlying CUB-200 is too specific for general LLMs.

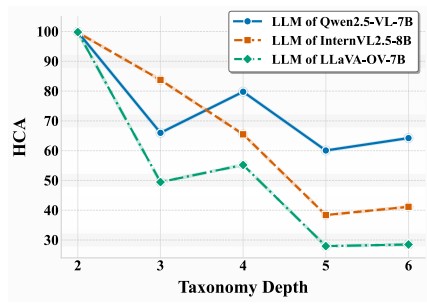

Figure 4: Text HCA of different VLLMs' LLMs over the iNat21-Plant taxonomies of various depths.

However, Table 3 further reveals that the LLMs also cannot perform well on ImgNet's general taxonomies. Besides, we progressively simplify our QA tasks by chopping the iNat21-Plant taxonomy level by level. Figure 4 plots the (text) HCA results, which increase as the taxonomy becomes shallower (and, correspondingly, the leaf nodes are less fine-grained). Still, they are below 90% regardless of the taxonomies' depths. There are noticeable drops at Levels 3 and 5 for Qwen2.5-VL and LLaVA-OV's LLMs, implying that they pose more challenges than the other levels for the LLMs' hierarchical reasoning. These results are surprising to a large degree, given the recent success of LLMs over various benchmarks and domains [1, 50, 60, 33, 45].

**Correlation between (text) HCA and $\text{Acc}_{\text{leaf}}$-scaled (visual) HCA.** An LLM's low (text) HCA undoubtedly discounts its corresponding VLLM's hierarchical consistency on visual inputs. We can quantify this notion using Pearson's correlation coefficient. Since the (text) HCA's corresponding leaf-level accuracy is 100% — we replaced images with their ground-truth leaf labels when making the text QA tasks, we normalize (visual) HCA by $1/\text{Acc}_{\text{leaf}}$. The last column in Table 3 shows that the correlation between (text) HCA and $\text{Acc}_{\text{leaf}}$-scaled (visual) HCA is as high as 0.9116.

*A note about GPT-4o's (text) HCA.* The analyses above apply to only open-source VLLMs because we cannot separate LLMs from the proprietary GPT-4o. Unlike the open-source LLMs' low (text) HCA, GPT-4o's (text) HCA scores are as high as 98.81. Hence, the LLM part is not GPT-4o's bottleneck in hierarchical visual understanding; instead, there are other possible causes of GPT-4o's hierarchical inconsistency about the visual world.

### 3.3.2 Why Are LLMs Poor at Hierarchical *Text* Classification?

In what follows, we present some preliminary quests into why and where LLMs fail at the seemingly simple hierarchical four-choice text classification tasks. We rule out the vision-language tuning that anchors visual encoders to pretrained LLMs and conclude that the language decoders are responsible for LLMs' lack of taxonomy knowledge.

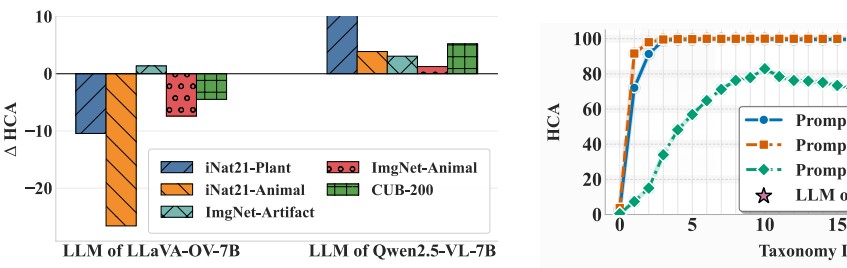

Figure 5: **Left**: (Text) HCA difference between vision-language-tuned LLMs and the original ones. **Right**: (Text) HCA of linearly probing different layers of Qwen-2.5-VL-7B's LLM on iNat21-Plant.

**Vision-Language Tuning Is *Not* the Reason.** Acute readers likely have noted that our previous LLM results are about the LLM parts of VLLMs, not the "true" standalone LLMs. Does the vision-language tuning, which is needed when one connects a visual encoder with an LLM, compromise LLMs and potentially induce catastrophic forgetting of taxonomy knowledge?

We answer this question by studying the original LLMs from which VLLMs are initialized, using the same text-only hierarchical classification setup described in Section 3.3. Figure 5 (Left) compares LLaVA-OV-7B and Qwen2.5-VL-7B's LLMs with their corresponding original LLMs. First of all, we see that the original LLMs are on par with or even worse than their vision-tuned counterparts, indicates that the standalone LLMs still lack a strong grasp of taxonomy knowledge. Interestingly, Qwen2.5-VL's LLM actually outperforms its original LLM on all taxonomies; in other words, the vision-language tuning actually enhances the LLM's (text) hierarchical consistency. In contrast, LLaVA-OV's vision-language tuning weakens the LLM's (text) HCA.

**LLMs Encode Hierarchical Structures Effectively but Cannot Decode Them Sufficiently.** Next, we shift attention to the LLM embeddings of the concepts in our taxonomies — if the embeddings do not provide sufficient hierarchical structural cues, there is little chance LLMs can decode them. To this end, we convert a taxonomy into language prompts of three variants:

> **Prompt 1:** `<leaf node label>` (e.g., Blue Jay) belongs to the `<hierarchy>` (e.g., Order) of `<ground truth>` (e.g., Passeriformes).
> **Prompt 2:** Given the `<leaf node label>`, what is its taxonomic classification at the `<hierarchy>` level? It belongs to `<ground truth>`.
> **Prompt 3:** Given the `<leaf node label>`, what is its taxonomic classification at the `<hierarchy>` level?

We then train a linear classifier for each taxonomy level to probe the average embedding of the language tokens in every layer of an LLM. Figure 5 (Right) summarizes the (text) HCA results of Qwen2.5-VL-7B's LLM on iNat21-Plant: The text embeddings give rise to highly hierarchically consistent linear probes. Especially for Prompt 3, with the ground-truth hierarchy labels withheld, the linear probes that receive only the leaf node embeddings can still achieve near-perfect hierarchical consistency in the LLM's deeper layers. In other words, the specialized linear probes can decode the taxonomy knowledge significantly better than the general-purpose LLM.

**LLMs' Hierarchical Orthogonality Does Not Guarantee Hierarchical Consistency.** Park et al. [42] recently predicted that LLMs represent hierarchical relations orthogonally in the representation space, e.g., `animal` is orthogonal to `bird−mammal`. They validated the prediction using Gemma [51] and LLaMA [19], and we further verify it in Figure 6 using both the original Qwen2.5-7B and the one after vision-language tuning. This pleasant geometric interpretation is, unfortunately, shadowed by the poor performance of Gemma and Qwen2.5-7B on our taxonomy QA tasks — we report the Gemma results in Appendix C. We argue that more fine-grained analyses of the LLM representation are required to establish a relationship between LLMs' hierarchical consistency and geometry.

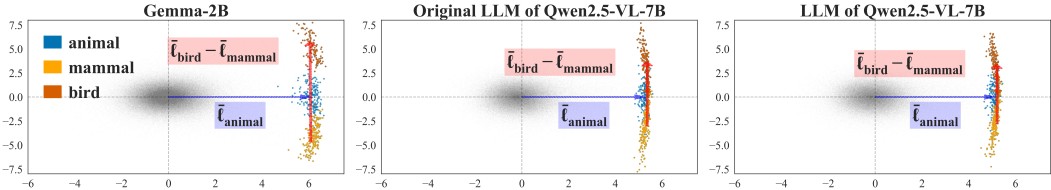

Figure 6: Hierarchical semantics are encoded as orthogonality in different LLMs' representation spaces (figures drawn following [42]).

## 4 LLMs Gain More Hierarchical Consistency than VLLMs from Finetuning

Could we improve the VLLMs' hierarchical visual understanding capabilities via finetuning using our VQA tasks built upon taxonomies? Likely, no, because LLMs are the bottleneck: The LLMs' hierarchical consistency over text-only tasks is so bad (Table 3) that we conjecture this shortcoming can only be fixed in the pretraining stage rather than the "tail patching" finetuning stage.

Still, the following presents some LoRA-finetuning [22] experiments with Qwen2.5-VL-7B, the best-performing 7B VLLM in our previous experiments, mainly for two reasons. One is to see how much finetuning could help, even though we believe pretraining instead of finetuning should be the rescue to VLLMs' hierarchical inconsistency. The other is further to investigate the interplay between VLLMs and their LLMs — interestingly, our results reaffirm that LLMs are the bottleneck for VLLMs' hierarchical visual understanding because LLMs' performance gain from the finetuning upper-bounds VLLMs'. Our finetuning data consists of VQA tasks constructed from iNat21-Plant's training set, covering 3,771 species nodes in the taxonomy instead of the full 4,271 species nodes. We then evaluate the finetuned model's improvement on iNat21-Plant, its generalization to other hierarchical visual understanding datasets, and how well it maintains the general vision-language capabilities. Please see Appendix D for more details on the training.

**Results and Discussion.** Tables 4 shows that finetuning Qwen2.5-VL using the VQA tasks that partially cover the iNat21-Plant taxonomy delivers improvements on both iNat21-Plant and other datasets. On iNat21-Plant, HCA rises from 17.67 to 29.34 (+11.67 absolute gain), while $\text{Acc}_{\text{leaf}}$

Table 4: (Visual) HCA and $\text{Acc}_{\text{leaf}}$ of Qwen2.5-VL-7B before and after the LoRA-finetuning.

| Model | iNat21-Animal | | iNat21-Plant | | ImgNet-Animal | | CUB-200 | |
|---|---|---|---|---|---|---|---|---|
| | HCA | $\text{Acc}_{\text{leaf}}$ | HCA | $\text{Acc}_{\text{leaf}}$ | HCA | $\text{Acc}_{\text{leaf}}$ | HCA | $\text{Acc}_{\text{leaf}}$ |
| Qwen2.5-VL-7B | 19.43 | 41.33 | 17.67 | 41.61 | 56.00 | 80.01 | 43.76 | 65.50 |
| Qwen2.5-VL-7B (LoRA) | 23.38 | 45.00 | 29.34 | 47.66 | 58.62 | 80.28 | 46.17 | 67.12 |
| $\Delta$ | +3.95 | +3.67 | +11.67 | +6.05 | +2.62 | +0.27 | +2.41 | +1.62 |

Table 5: (Text) HCA of the LLM of Qwen2.5-VL-7B before and after the LoRA-finetuning.

| Model | iNat21-Animal | iNat21-Plant | ImgNet-Animal | CUB-200 |
|---|---|---|---|---|
| LLM of Qwen2.5-VL-7B | 52.08 | 64.21 | 68.14 | 63.86 |
| LLM of Qwen2.5-VL-7B (LoRA) | 65.63 | 84.87 | 72.39 | 66.15 |
| $\Delta$ | +13.55 | +20.66 | +4.25 | +2.29 |

gains 6.05. The HCA on ImageNet-Animal increases from 56.00 to 58.62 and on CUB-200 from 43.76 to 46.17. More interestingly, Table 5 indicates that the LLM's (text) HCA increases more from the finetuning than Qwen2.5-VL's (visual) HCA (e.g., 20.66 vs. 11.67 on iNat21-Plant and 4.25 vs. 2.62 on ImgNet-Animal). To some extent, this finding reaffirms that LLMs are the bottleneck of VLLMs' hierarchical visual understanding, and one has to improve LLMs' (text) taxonomy knowledge to boost VLLMs' (visual) hierarchical consistency. Besides, our results demonstrate that vision-language training can benefit both VLLMs and their LLMs, aligning with some recent advocates for improving LLMs using multimodal data beyond language only [31, 52]. Appendix D reports more results and discussion, including that the finetuned model does not lose its general capability tested on MME [16], MMBench [36], and SEED-Bench [30].

## 5 Related Work

Hierarchical classification [47, 25] enables many applications. It is vital for a comprehensive understanding of the visual world [61, 43, 65, 48, 7, 44] and many language concepts [70, 56, 71, 21]. Several recent studies have revisited this longstanding problem and shown that CLIP-style [46] models lack consistency across taxonomic levels [58, 18]. Wu et al. [58] evaluate CLIP under multiple levels of semantic granularity and introduce a hierarchy-consistent prompt tuning method. Pal et al. [40] enhance CLIP's hierarchical representations by embedding them to a hyperbolic space. Xia et al. [59] further extend this direction by incorporating graph-based representation learning. Novack et al. [38] use hierarchical information to improve zero-shot classification accuracy. Zhang et al. [68] first identified the limitations of current VLLMs in fine-grained image classification. Building on this, Liu et al. [35] further assess a broader range of VLLMs. He et al. [20] point out a potential cause, the scarcity of image class names in pretraining. Beyond closed-set evaluation [63, 17], Conti et al. [12] benchmark VLLMs' open-world classification, while Snæbjarnarson et al. [49] propose to evaluate VLLMs' open-set predictions using a taxonomic similarity rather than exact string matching. However, to the best of our knowledge, no prior work has examined VLLMs under the hierarchical visual understanding context.

## 6 Conclusion

This work presents a systematic evaluation of state-of-the-art VLLMs's hierarchical visual understanding performance. We find that both open-source VLLMs and the proprietary GPT-4o give rise to low hierarchical consistency over six taxonomies of visual concepts. Probing results reveal that the visual and text embeddings carry rich hierarchical and discriminative cues, whereas the LLMs fail to decode them, implying LLMs are the bottleneck. Finetuning on hierarchical VQA tasks improves VLLMs' hierarchical consistency on visual inputs while preserving their performance on general VQA tasks. Intriguingly, the finetuning benefits the LLMs (text) hierarchical consistency more than the corresponding VLLM's (visual) hierarchical measure. Ingesting the taxonomy-knowledge gap to LLMs, likely during pretraining rather than post-hoc patching, is a promising path toward VLLMs that reason coherently across different levels of semantic granularity about the visual world.

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
