# OpenReview forum: "Vision LLMs Are Bad at Hierarchical Visual Understanding, and LLMs Are the Bottleneck"
_NeurIPS.cc/2025/Conference — Submitted to NeurIPS 2025_

### Official Review · Reviewer_71bs · 2025-06-08

**Clarity:** 2
**Significance:** 2
**Originality:** 3
**Rating:** 3
**Confidence:** 3

**Summary:**

This work studies the hierarchical reasoning capabilities of vision-language models, primarily vision LLMs (VLMMs), finding that they have poor performance on tasks requiring reasoning over taxonomies. Various tests including probing experiments suggest that the bottleneck stems from their LLM components, and examine the effect of fine-tuning VLLMs on hierarchical data.

**Questions:**

What is the baseline human performance for the tasks in Section 2?

Have you tested SOTA uni-modal LLMs (rather than LLM components of VLLMs) for similar hierarchical understanding?

What possible applications would be enabled by improving hierarchical reasoning in VLLMs?

**Ethical Concerns:**

["NO or VERY MINOR ethics concerns only"]

**Final Justification:**

Several of the concerns have been addressed. However, concerns regarding human baseline results, zero-shot vs. supervised performance gaps, and writing and stylistic issues are still relevant.

**Limitations:**

I appreciate the limitations section in Supp. E. However, it is missing some limitations raised above, such as testing only on taxonomies with short labels (as discussed above).

**Quality:**

2

**Strengths And Weaknesses:**

**Strengths:**

Understanding the hierarchical reasoning capacity of SOTA VLMs is important, and does seem to address a gap in the existing literature which mostly focuses on hierarchical understanding in CLIP-style discriminative models.

**Weaknesses:**

While the chosen research question is interesting and addresses an important gap in the literature, there are experimental and stylistic issues which should be addressed:

*Experimental issues:*

While the paper claims that “Vision LLMs are Bad at Hierarchical Visual Understanding” (including in the title), the tests in Section 2 are mostly on biological taxonomies, as well as a few other taxonomy types (Food, WordNet) which are limited to words or short labels and have a large proportion of obscure or technical terms. No human baseline is given, but it is doubtful that humans would typically succeed in identifying e.g. whether “Anseriforme” correctly describes an animal. Prior work [1, 2] has found foundation VLMs can have hierarchical understanding, particularly of longer, more natural caption texts, but this is not tested (e.g. against the HierarCaps benchmark of [2]).

The claim that visual embeddings are not the bottleneck to hierarchical understanding (Sec 3.2) uses linear probing, but this is a supervised method rather than the zero-shot classification used in Section 2. This gives a significant advantage to learning the label hierarchy and thus I do not find this experiment to conclusively show that visual embeddings inherently encode this hierarchical knowledge better than LMMs.

The findings in Section 4 seem to be of limited significance since it is expected that fine-tuning on a task with its particular label set will outperform zero-shot classification. In addition, if the claim on L282-3 is included, it should be supported by experimental evidence.

*Stylistic issues:*

The paper’s organization and writing style distract from its contents and would benefit from revision. For example, long section titles (e.g. “2 VLLMs Lack Hierarchical Consistency in Visual Understanding”) state conclusions before evidence is presented, rather than using a more standard naming convention, and the first lines of the abstract refer only to LLMs while the paper focuses on multi-modal VLMMs. The internal structure and tone of the paper’s text can also be refined for clarity.

There are also missing references and discussion of existing works on hierarchical understanding in foundation vision-language models [1, 2].

[1] Desai et al. Hyperbolic image-text representations. ICML 2023

[2] Alper et al. Emergent Visual-Semantic Hierarchies in Image-Text Representations. ECCV 2024

---

> ### Author Rebuttal · Authors · 2025-07-30
>
> Thank you for your thoughtful comments and valuable feedback. Please find our detailed responses to the weaknesses and questions below.
>
> > No human baseline is given, but it is doubtful that humans would typically succeed in identifying e.g. whether “Anseriforme” correctly describes an animal.
>
> We thank the reviewer for this thoughtful comment. We agree that humans are unlikely to perform well on benchmarks such as CUB-200/INat-21 Plant/INat-21 Animal without expert knowledge. Recruiting domain experts to cover the full taxonomy would also be highly impractical.
>
> We believe this limitation underscores a key challenge in hierarchical visual understanding: the knowledge required spans multiple domains, and even human experts are typically specialized in only narrow areas. This highlights an important distinction: while domain-specific models may perform well within limited scopes, VLLMs are expected to generalize across domains. We argue that excelling at hierarchical understanding across such diverse semantic spaces is a critical capability for VLLMs as they aspire toward general-purpose intelligence.
>
>
> > Prior work [1, 2] has found that foundation VLMs can have hierarchical understanding, particularly of longer, more natural caption texts, but this is not tested (e.g. against the HierarCaps benchmark of [2]).
>
> Desai et al. [1] actually identify the lack of explicit hierarchical modeling in CLIP and introduce a contrastive learning framework that embeds images and text into a shared hyperbolic space to better capture hierarchical semantics.
>
> We evaluate HierarCaps [2] under our experimental settings, and the results are presented below.
>
> | Model        | Level 4 | Level 3 | Level 2| Level 1 | HCA (All)  | HCA (Last two levels) |
> |------------------|--------:|--------:|--------:|--------:|--------:| --------:|
> | OpenCIP  |  68.80  |  50.70  |  28.40  | 17.10   | 5.70   |  45.30 |
> | SigLIP  |   72.90 |  52.50  |  26.80  | 15.5   |  5.70  | 49.20  |
> | LLaVA-OV-7B  |  78.60  | 60.70   |  32.60  |  22.20  |  9.10  | 55.60  |
> | Qwen2.5-VL-7B   |  77.10  | 58.40   |  30.50  |   21.10 |  7.10  | 54.00  |
>
> From the table, we observe that both CLIPs and VLLMs have poor hierarchical consistency (see the low HCA scores) and are especially worse on abstract and shorter captions at higher levels of the hierarchy (i.e., the first two levels). This is likely due to the noisy nature of the top-level labels in the HierarCaps dataset, where each image can be associated with multiple coarse-grained captions. For example, the image with the caption “A table with three plates with food and a man and a woman both holding utensil near plates” can reasonably be classified under both “persons” and “table” categories at the first layer of the hierarchy.
>
> We also compute the HCA for the last two levels (Level 3 and Level 4), which contain more concrete and longer captions. However, there remains a noticeable gap between the HCA and the leaf node accuracy (Level 4). In many cases, the model can correctly classify at level 4 while still struggling at level 3, suggesting that the model does not fully understand the hierarchical relationship across longer textual contexts.
>
> [1] Desai K, Nickel M, Rajpurohit T, et al. Hyperbolic image-text representations[C]//International Conference on Machine Learning. PMLR, 2023: 7694-7731.
>
> [2]  Alper M, Averbuch-Elor H. Emergent visual-semantic hierarchies in image-text representations[C]//European Conference on Computer Vision. Cham: Springer Nature Switzerland, 2024: 220-238.
>
> > The claim that visual embeddings are not the bottleneck to hierarchical understanding (Sec 3.2) uses linear probing, but this is a supervised method rather than the zero-shot classification used in Section 2. This gives a significant advantage to learning the label hierarchy and thus I do not find this experiment to conclusively show that visual embeddings inherently encode this hierarchical knowledge better than LMMs.
>
> Thank you for raising this important point. We would like to clarify the intention of our probing experiment, as discussed in Lines 183–190 and 200–202 of the paper. Our goal in Section 3.2 is to investigate whether the visual embeddings contain hierarchical information, independent of whether the LLM can decode it in a zero-shot manner.
>
> To do this, we use linear probing as a standard and interpretable diagnostic tool. It allows us to ask a focused question: Is the hierarchy easily decodable from the visual embeddings? Our results in Figure 3 indicate that the visual embeddings do carry rich and consistent hierarchical and discriminative cues, but the LLM does not effectively utilize these cues in downstream VQA tasks.
>
> In short, the probing experiment highlights that there is no representational bottleneck in visual embeddings as the visual tokens remain discriminative and structurally rich throughout different LLM layers.
>
> > The findings in Section 4 seem to be of limited significance since it is expected that fine-tuning on a task with its particular label set will outperform zero-shot classification. In addition, if the claim on L282-3 is included, it should be supported by experimental evidence.
>
>
> Thank you for the comment. We agree that the improvement on the fine-tuned task itself is expected, and we have acknowledged this in our paper as stated in Lines 280–281 that fine-tuning is likely not the solution. However, the significant finding lies elsewhere (Lines 299-303): a comparison between the results from Table 4 and 5 further reinforces our main point that the LLM is the bottleneck in hierarchical visual understanding.
>
>
>
> Regarding Lines 282–283, we’d like to clarify that this is not a claim but a conjecture. We clearly express that we are unable to verify it through large-scale pretraining in the Limitation, and we offer it only as a possible explanation to motivate future work.
>
>
> > The paper’s organization and writing style distract from its contents and would benefit from revision. The internal structure and tone of the paper’s text can also be refined for clarity.
>
>
> We sincerely thank the reviewer for the suggestions on improving the writing and organization. We acknowledge that some section titles may be longer than usual and contain conclusive phrasing. This was a deliberate stylistic choice, as the paper is analysis-driven, and we wanted to convey the key findings up front to guide the reader through the investigation. That said, we understand the importance of clarity and standard structure, and we will revise the writing to further improve on those aspects.
>
> **Questions:**
>
> > What is the baseline human performance for the tasks in Section 2?
>
> Please see the answer to weakness 1.
>
> > Have you tested SOTA uni-modal LLMs (rather than LLM components of VLLMs) for similar hierarchical understanding?
>
> Yes, we evaluate the uni-modal LLMs in Section 3.3.2, where we test the original language models from which the VLLMs are initialized, using the same text-only hierarchical classification setup. Specifically, we include:
> * Qwen2.5-7B-Instruct (used in Qwen2.5-VL),
> * Qwen2-7B-Instruct  (used in LLaVA-OV),
> * InternLM2.5-7B-Chat (used in InternVL2.5).
>
> Interestingly, we find that the original LLMs perform on par with or in some cases worse than their vision-tuned counterparts, as shown in Figure 5 Left, indicating that standalone LLMs still lack a strong grasp of taxonomic knowledge.
>
> Notably, the LLM component of Qwen2.5-VL consistently outperforms the original Qwen2.5 across all taxonomies. This suggests that vision-language tuning can enhance the LLM’s hierarchical consistency even in text-only settings. In contrast, LLaVA-OV’s vision-language tuning weakens the LLM’s (text) HCA.
> > What possible applications would be enabled by improving hierarchical reasoning in VLLMs?
>
> Enhancing hierarchical consistency is foundational to robust, generalizable, and explainable visual intelligence.  We list several possible applications below.
> * Visual question answering and multi-turn reasoning, hierarchically structured outputs support logical follow-up interactions. For instance, a user might ask, “What animal is this?” followed by “What kind of mammal?” and “What breed?”, all of which require a model to maintain taxonomic coherence across its responses.
> * Ecological monitoring and biodiversity surveys. In open-world or long-tail settings, hierarchical predictions allow fallback to broader yet correct labels (e.g., Animal → Vertebrate → Fish), which remain informative when fine-grained classes are unknown.
> * Taxonomy-aware image retrieval and visual search. Hierarchically aligned predictions enable users to query or filter results at different semantic levels, leading to more structured and relevant search experiences, especially in large-scale visual databases.
> * Educational and scientific AI tools such as species identification apps, interactive learning platforms, or museum companions—benefit from being able to explain visual content through layered taxonomy paths, which fosters better user understanding.

---

> > ### Comment · Reviewer_71bs · 2025-08-01
> >
> > Thank you for your response.
> >
> > > Recruiting domain experts to cover the full taxonomy would also be highly impractical
> >
> > Acknowledged, though it would be beneficial to have a layman human baseline as many hierarchical labels are more solvable for untrained humans (e.g. identifying invertebrates, mammals, ...) and foundation VLMs trained on Internet captions might better reflect this common knowledge.
> >
> > > Desai et al. [1]
> >
> > My intent here was that a foundation VLM such as hyperbolic CLIP is not explicitly trained on hierarchies but was found to better encode hierarchical knowledge.
> >
> > > From the table, we observe that both CLIPs and VLLMs have poor hierarchical consistency (see the low HCA scores) and are especially worse on abstract and shorter captions at higher levels of the hierarchy (i.e., the first two levels).
> >
> > Is this consistent with the findings of [2] that these models have emergent hierarchical understanding, reflected in various other metrics?
> >
> > > the probing experiment highlights that there is no representational bottleneck in visual embeddings as the visual tokens remain discriminative and structurally rich throughout different LLM layers
> >
> > I agree that this is a good demonstration that this hierarchical knowledge is effectively discernible in visual embedding space.
> > However, I'm not sure it is remarkable (L194) that zero-shot decoding performs worse at predicting the particular label hierarchy used in these datasets, as linear probing is supervised and is expected to better correspond to the dataset format. A similar effect is observable in standard image classification tasks where linear probes on standard vision embeddings will often perform better than zero-shot VLM classification when evaluating against a particular label set. Maybe the presentation can be toned down as this seems like the expected outcome.

---

> ### Author Response · Authors · 2025-08-02
>
> Thank you for your prompt response. Please find our reply to your comments below.
>
> >Acknowledged, though it would be beneficial to have a layman human baseline as many hierarchical labels are more solvable for untrained humans (e.g. identifying invertebrates, mammals, ...) and foundation VLMs trained on Internet captions might better reflect this common knowledge.
>
>
> Thank you for the suggestion. We agree that including a layman human baseline would be valuable, especially for assessing performance on more intuitive labels such as “invertebrates” or “mammals,” which may be solvable by non-experts.
>
>
> However, due to the scale of our benchmark (approximately 1 million QA pairs), conducting a thorough human evaluation would require substantial labor and resources beyond our current capacity. While we could randomly sample 1/10 of the images from each leaf node to construct a subset for human evaluation, the subset would still contain about 100k QA pairs. As a result, we are currently unable to provide this baseline.
>
>
> That said, we recognize the importance of establishing a human baseline and will make our benchmark publicly available. We hope this will encourage community contributions toward building a human baseline in future work.
>
>
> > Desai et al. [1]. My intent here was that a foundation VLM such as hyperbolic CLIP is not explicitly trained on hierarchies but was found to better encode hierarchical knowledge.
>
>
> Sorry for the misunderstanding regarding that question, and thank you for your clarification.
>
>
> Hyperbolic space offers a natural advantage for representing data with inherent hierarchical or tree-like structures. The number of nodes increases exponentially with depth in hierarchies, progressing from a few coarse-grained to many fine-grained categories. While the volume of a ball in Euclidean space grows only polynomially with its radius, hyperbolic space, characterized by constant negative curvature, exhibits exponential volume growth, closely mirroring the expansion patterns in hierarchical data. This geometric property allows models to more effectively encode hierarchical relationships even without explicit supervision from hierarchical labels compared to Euclidean representations.
>
> >Is this consistent with the findings of [2] that these models have emergent hierarchical understanding, reflected in various other metrics?
>
>
> To some extent, this aligns with the findings of [2], which show that CLIP models are capable of image-text matching not only at the leaf level but also at more abstract upper levels. Similarly, our results indicate that, in addition to the leaf level (level 4), CLIPs demonstrate reasonable performance across the three higher levels, with performance gradually decreasing as the hierarchy becomes more abstract. However, the visual hierarchical consistency (as measured by HCA) is low in our evaluation. This aspect is not explored in [2], which only reports high hierarchical consistency for text embeddings ($\tau_d$) in Table 1 of their paper.
>
>
> >I agree that this is a good demonstration that this hierarchical knowledge is effectively discernible in visual embedding space. However, I'm not sure it is remarkable (L194) that zero-shot decoding performs worse at predicting the particular label hierarchy used in these datasets, as linear probing is supervised and is expected to better correspond to the dataset format. A similar effect is observable in standard image classification tasks where linear probes on standard vision embeddings will often perform better than zero-shot VLM classification when evaluating against a particular label set. Maybe the presentation can be toned down as this seems like the expected outcome.
>
>
> Thank you for the insightful comment.
>
> We agree that the performance of visual probing is expected to surpass that of zero-shot decoding. As shown in Figure 3 of the main paper, on the CUB-200 dataset, the HCA achieved by visual probing exceeds that of zero-shot decoding by nearly 40%. While we believe this is a substantial performance gap, we acknowledge that describing it as "remarkable" may be overly subjective. We will revise the corresponding statement to a more neutral and moderate tone.

---

> ### Comment · Reviewer_71bs · 2025-08-04
>
> Thank you for your response. This addresses several of my concerns, although others are not fully addressed (human performance, zero-shot vs. supervised performance and presentation, writing issues in paper). I will update my score accordingly.

---

> > ### Author Response · Authors · 2025-08-04
> > **Follow up with a clarification question**
> >
> > Dear Reviewer 71bs: Thank you for the further review of our paper and rebuttal. We wonder if you could give us the opportunity to resolve those unaddressed questions, for example,
> >
> > + Human performance: Is it reasonable to have human performance on 1% of the data? It's feasible if so. It's too expensive to us to study more than that.
> > + Zero-shot vs supervised & presentation: Could you clarify which parts of the answers could be improved?
> >
> > Sincere & Thanks,
> > Authors

---

> ### Comment · Reviewer_71bs · 2025-08-06
>
> Thank you. To clarify, regarding human performance I think it would help to have any investigation into whether poor performance on the given taxonomies is due to the abundance of obscure labels which humans themselves struggle on (and which are likely under-represented in Internet image caption data).
>
> Regarding the zero-shot vs. supervised gap, my concern is that it is expected to have a significant gap in performance with methods that are explicitly given access to the hierarchy labels and format as training data, and I'm not sure that this reflects a novel or meaningful finding. This seems to undermine the central claim that LLMs bottleneck hierarchical visual understanding.
>
> I also still have significant issues with the paper's writing, which I believe needs moderate editing for tone and clarity.

---

> > ### Author Response · Authors · 2025-08-06
> >
> > Thank you so much for the further feedback, which will significantly help us improve the paper.
> >
> > RE human performance: We are running a small-scale human study among ourselves, and we find that the "obscure labels" are not a concern as we can understand them by referring to explanation in layman's language. Instead, the primary challenge is the lack of knowledge about the (e.g., biological) taxonomies.
> >
> > RE Zero-shot vs. supervised: Thank you for the insightful discussion about this point. If my understanding is correct, you are suggesting that the visual embeddings have dual roles: 1) Encoding of hierarchical knowledge and 2) Compatibility with LLMs. The linear probing experiments can only confirm 1), but not 2). Essentially, a high-quality visual embedding for VLMs should be good in **the context of LLMs**--- they should both encode sufficient visual cues and be compatible, decode-able by LLMs. Please confirm if our understanding is correct or not; if so, we will make this point clearer in the paper.
> >
> > RE Writing: We understand that the reviewer has reservations about the writing style. We sincerely request the reviewer's accommodation for different writing styles if they can deliver clarity and scientific reproducibility.

---

> > > ### Comment · Area_Chair_W749 · 2025-08-09
> > > **Please post feedback (if you have any)**
> > >
> > > Dear reviewer, as the discussion period is approaching ending, we would be appreciated if you can post feedback at this point especially where Authors ask for such discussions. For your comments that arrive late - Authors may have insufficient time now to address them in depth.

---

> > > ### Comment · Reviewer_71bs · 2025-08-09
> > >
> > > Thank you for your response.
> > >
> > > Regarding the request for clarification - I do not fully agree with the claim in (1) that linear probing shows that visual embeddings encode hierarchical knowledge, because the linear probe itself is supervised, which provides the classifier information about the label space itself. This contrasts with zero-shot probing methods which demonstrate that knowledge is inherently encoded in a representation.

---

> > > > ### Author Response · Authors · 2025-08-09
> > > >
> > > > Thank you, Reviewer 71bs, for the further clarification of your question.
> > > >
> > > > > … because the linear probe itself is supervised, which provides the classifier information about the label space itself.
> > > >
> > > > We gently remind the reviewer of a subtle mistake in the reasoning. If we replace visual embeddings with random noise, the linear probing would lead to chance results even though it “provides the classifier information about the label space itself.”
> > > >
> > > > > … zero-shot probing methods which demonstrate that knowledge is inherently encoded in a representation.
> > > >
> > > > We respectfully clarify that the point is *not* to compare linear probing against zero-shot probing for their performance per se. Rather, the linear probing (together with the other analyses in the paper) is meant to pinpoint the cause to the poor zero-shot hierarchical consistency.
> > > >
> > > > Finally, just in case, we reiterate the following mentioned earlier: The linear probing uses the labels per level in a taxonomy as one-hot vectors (with no semantics or explicit hierarchical constraints/regularization), so the surprisingly high hierarchical consistency of the linear probing largely attributes to the visual embeddings’ quality.

---

### Official Review · Reviewer_7doG · 2025-06-15

**Clarity:** 4
**Significance:** 4
**Originality:** 4
**Rating:** 6
**Confidence:** 4

**Summary:**

This paper studies whether VLLM systems can answer questions about  hierarchical (IS-A) relations. The authors create a large set of VQA tasks drawn from hierarchies and datasets, and then test performance.  They find performance to be poor, and then investigate the locus of the problem, for example by probing the internal representations of the VLLM at various stages.  They find that the culprit is the LLM, suggesting that LLMs lack hierarchial knowledge about the visual world.

**Questions:**

1.  The authors should fix the weakness noted above

2. The authors could increase their discussion of why there are performance differences on different datasets. For example, the difference in performance between ImageNet-Animal,  and Image-Net-Artifiact is likely that ImageNet-Animal, the leaf node is often the basic-level category, but in ImageNet-Artifact, the leaf node is often subordinate-level, and the basic category is one of the intermediate nodes, making the task much harder.

3. The authors used zero-shot prompting for section 3.3.  Would the results change and performance increase if few-shot prompting was used instead?

4. The authors seem to have missed much of the prior literature on investigating hierarchies in text language models. It's not that extensive, and a lot of it is old and uses BERT, but still. Some papers:

https://aclanthology.org/2023.findings-ijcnlp.12.pdf

https://aclanthology.org/2022.acl-short.11/

https://arxiv.org/pdf/2302.06761

**Ethical Concerns:**

["NO or VERY MINOR ethics concerns only"]

**Final Justification:**

Still a strong paper and author responses confirm that.

**Limitations:**

yes

**Paper Formatting Concerns:**

I didn't understand line 87-88; perhaps it was confusingly worded?

line 206: the word "influential" is unnecessary

linei 207: typo Souce for Source

line 267 "Especially for Prompt 3" do you mean "Even for Prompt 3"?

Line 263 I didn't understand what the LM was exactly given for the 3 variants, in particular Prompts 1/2 vs Prompt 3.  Prompt 1/2 are not questions. Does that mean the LM for prompt 1 was given everything up to the word "of" and then greedy decoding was taken to generate the ground truth?

**Quality:**

4

**Strengths And Weaknesses:**

Strengths:

1. The paper asks an important question (hierarchical visual knowledge in VLLMs), and comes up with an interesting and important finding (a lack of this knowledge in current VLLM models)  and  a lovely causal analysis (localizing the cause of the flaw to the LLM component)

2. The experiments are comprehensive and convincing.

3. The paper is beautifully and clearly written and I enjoyed reading

Weaknesses:

1. The paper overstates its claims; the results that the LLM is the bottleneck hold only for the open-source models studied. As the paper makes clear at line 240, the results do not hold for GPT-4o, but only for the open models. Of course since GPT-4o is closed, it's reasonable that the authors can't say why GPT-4o can do the text hierarchy task but not the vision hierarchy task.  Still, this should be more clearly explained throughout the paper. Currently the sentence at line 165 and the sentence at line 240 are the only place where I was given to understand as a reader that the paper's causal conclusions don't hold for the closed models (and the sentence at line 165 and 240 don't seem completely consistent with each other in their implications).  The abstract was careful to use the word "many" in "many state-of-the-art LLMs", which is a good start, but the reader could legitimately feel deceived. The paper is a wonderful paper, so adding more careful and thoughtful caveats and clear definitions in the right places would help a lot without lessening the findings.

---

> ### Author Rebuttal · Authors · 2025-07-30
>
> Thank you so much for thoroughly understanding and positively evaluating our work. We are glad to hear that you enjoyed our work. Below, we respond to your insightful suggestions to further robustify our claims.
>
> > …the LLM is the bottleneck hold only for the open-source models studied. …it's reasonable that the authors can't say why GPT-4o can do the text hierarchy task but not the vision hierarchy task. …Adding more careful and thoughtful caveats and clear definitions in the right places would help a lot without lessening the findings.
>
> We greatly appreciate the suggestion! We will revise the paper to present our claims more clearly and rigorously as follows:
>
> * We will modify the title to “Vision LLMs Are Bad at Hierarchical Visual Understanding, and LLMs Are the Bottleneck for Open-source Vision LLMs”.
> * We will move Lines 165-167 (We believe this conclusion is true for open-source VLLMs, but we urge readers not to extrapolate it to proprietary LLMs because we could not probe their intermediate embeddings.) to the end of the Introduction.
> * We will add the keyword “open-source” to the first and last sentences of our abstract to improve clarity and rigor. Specifically, we will revise the abstract as follows:
>
>   **First sentence:** This paper reveals that many open-source state-of-the-art large language models (LLMs) lack hierarchical knowledge about our visual world, unaware of even well-established biology taxonomies.
>
>   **Last sentence:** We conjecture that one cannot make open-source vision LLMs understand visual concepts fully hierarchical until LLMs possess corresponding taxonomy knowledge.
>
>    If space permits, we will also incorporate “GPT-4o can do the text hierarchy task but not the vision hierarchy task” from your comments to the abstract (with minor updates to some wording).
>
> * We will revise line 240 as follows: Hence, we conjecture that the LLM part is likely not GPT-4o’s bottleneck in hierarchical visual understanding; instead, there are other possible causes of GPT-4o’s hierarchical inconsistency about the visual world.
>
> **Questions:**
>
> > The authors could increase their discussion of why there are performance differences on different datasets.
>
> Thank you for your valuable suggestion. We will add your insight about the ImageNet Artifact vs. Animal along with the following to the revised paper.
> Among all datasets, CUB-200 exhibits the smallest gap between HCA and leaf-level accuracy, which can be attributed to its shallow hierarchy (only four levels vs. six levels in iNat21-Plant and iNat21-Animal), making the task relatively simple compared to other datasets.
>
> ImgNet-Animal and ImgNet-Artifact have the deepest hierarchies. However, their leaf nodes generally correspond to basic-level concepts, which makes the leaf-level classification easier for VLLMs, resulting in high leaf accuracy.
>
> ImgNet-Artifact is the most challenging dataset on which all models yield low hierarchical consistency scores, probably for two main reasons. (1) Unlike the well-defined biological hierarchies in iNat-21 and ImgNet-Animal, the intermediate nodes in WordNet, which were used to construct ImageNet, are relatively abstract and vague, making hierarchical discrimination difficult. (2) Unlike the animal images, images in the ImgNet-Artifact dataset often contain multiple objects of different classes. When queried about higher-level categories, the model may mistakenly associate the question with another non-central object in the scene.
>
> >The authors used zero-shot prompting for section 3.3. Would the results change and performance increase if few-shot prompting was used instead?
>
> We conducted additional experiments on the CUB-200 dataset using Qwen2.5-VL-7B with few-shot prompting ranging from 1 to 5 shots. We use level-specific QA pairs as few-shot examples to evaluate performance at each hierarchy level. An example is provided as follows:
>
> ```
> Based on taxonomy, where does the <leaf label> (e.g., Black-footed Albatross) fall in terms of <level> (e.g., order)?
> A. <Ground Truth> (e.g., procellariiformes)
> B. <Similar Choice> (e.g., apodiformes)
> C. <Similar Choice> (e.g., podicipediformes)
> D. <Similar Choice> (e.g., pelecaniformes)
> Answer with the option's letter from the given choices directly.
> Answer: A
> ```
> The results are presented in the table below.
>
> | #Few-shot | 0 | 1 | 2 | 3 | 4 | 5 |
> |-------------------|---|---|---|---|---|---|
> |  HCA   |66.26| 64.83  | 65.57  | 65.50  | 66.10  | 65.46 |
>
> Interestingly, we observe no performance improvement across different numbers of few-shot examples.
>
> In addition, in a similar spirit to what you suggested, we explored another prompting experiment beyond zero-shot in Section C.1.3, and the results are in Table 15. Specifically, we include a text-only evaluation where each prompt is provided with the full taxonomy as context. As shown in Table 15, the text-only HCA reaches 74.82%, with an 8.56% gain, but still falls short of what we would expect from an LLM that already has access to the explicit answer for each question. This supports the same conclusion we drew from the few-shot experiments.
>
> > The authors seem to have missed much of the prior literature on investigating hierarchies in text language models.
>
>
> We appreciate the reviewer’s suggestion and acknowledge that we did not sufficiently discuss earlier work on hierarchical understanding in the pure language space. We will add a paragraph in the 'Related Works' section.
>
> Hierarchical understanding has been extensively explored in the language domain. Nikishina et al. [1] provide a comprehensive analysis of transformer-based models for the task of hypernymy prediction, evaluating their ability to infer IS-A relations. Lin and Ng [2] investigate whether pre-trained BERT models capture the transitivity of IS-A relations in WordNet. To more rigorously assess such capabilities, He et al. [3] propose ONTOLAMA, a benchmark and evaluation framework targeting subsumption inference within ontologies. Similarly, Moskvoretskii et al. [4] evaluate the WordNet-based lexical-semantic reasoning ability of the LLaMA-2-7B model through the TaxoLLaMA framework. Beyond analysis, a parallel line of work [5,6,7] has focused on injecting hierarchical structure into language models to improve their taxonomy-awareness.
>
> [1] Nikishina, Irina, et al. "Predicting terms in IS-a relations with pre-trained transformers." Findings of the Association for Computational Linguistics: IJCNLP-AACL 2023 (Findings). 2023.
>
> [2] Lin, Ruixi, and Hwee Tou Ng. "Does BERT know that the IS-a relation is transitive?." Proceedings of the 60th Annual Meeting of the Association for Computational Linguistics (Volume 2: Short Papers). 2022.
>
> [3] He Y, Chen J, Jimenez-Ruiz E, et al. Language model analysis for ontology subsumption inference[J]. arXiv preprint arXiv:2302.06761, 2023.
>
>
> [4] Moskvoretskii, Viktor, et al. "TaxoLLaMA: WordNet-based model for solving multiple lexical semantic tasks." arXiv preprint arXiv:2403.09207 (2024).
>
> [5] Zhou J, Ma C, Long D, et al. Hierarchy-aware global model for hierarchical text classification[C]//Proceedings of the 58th annual meeting of the association for computational linguistics. 2020: 1106-1117.
>
> [6] Wang Z, Wang P, Huang L, et al. Incorporating hierarchy into text encoder: a contrastive learning approach for hierarchical text classification[J]. arXiv preprint arXiv:2203.03825, 2022.
>
> [7] He Y, Yuan M, Chen J, et al. Language models as hierarchy encoders[J]. Advances in Neural Information Processing Systems, 2024, 37: 14690-14711.
>
> **Paper formatting:**
>
> > I didn't understand line 87-88; perhaps it was confusingly worded?
>
>
> We apologize for the confusion caused by this sentence and have rephrased it for clarity as follows:
>
>
> Interestingly, $\mathrm{Acc_{leaf}}$ upper-bounds $\mathrm{HCA}$ because correctly assigning a leaf label $y_L$ to an input $x$ contributes to $\mathrm{Acc_{leaf}}$, but correct leaf-level prediction does not increase $\mathrm{HCA}$ unless the model makes no mistake over all nodes in the path $(y_0,y_1,\cdots,y_L)$ connecting the leaf node to the root.
>
>
> > line 206: the word "influential" is unnecessary
>
>
> We will remove the word "influential".
>
>
> > linei 207: typo Souce for Source
>
>
> Thanks for pointing out the typo. Will revise it in the new version.
>
>
> > line 267 "Especially for Prompt 3" do you mean "Even for Prompt 3"?
>
>
> Yes, we mean “Even for Prompt 3” and will revise it.
>
>
> > Line 263 I didn't understand what the LM was exactly given for the 3 variants, in particular Prompts 1/2 vs Prompt 3. Prompt 1/2 are not questions. Does that mean the LM for prompt 1 was given everything up to the word "of" and then greedy decoding was taken to generate the ground truth?
>
>
> The prompts in line 263 are used to extract text embeddings from the LLM for linear probing. Specifically, each of the three prompts (Prompts 1, 2, and 3) is fed to the LLM in a single forward pass, and we extract the embeddings of the text tokens from each decoder layer. Since our focus there is on analyzing the internal representations of LLMs, we do not consider next-token prediction; we only use the token embeddings for probing.

---

> > ### Comment · Reviewer_7doG · 2025-08-04
> >
> > Thanks for all the clarifications and updated results! I'm happy with these and keep my score.

---

### Official Review · Reviewer_V1RC · 2025-07-01

**Clarity:** 4
**Significance:** 3
**Originality:** 3
**Rating:** 5
**Confidence:** 4

**Summary:**

The authors probe the hierarchical understanding of four visual LLMs, two CLIP-style models, and GPT-4o. To do so, they derived ~1 million visual question answering tasks from three biological datasets (the animal and planet split of iNat plus CUB-200), two splits of ImageNet (Animals and Artifact), and Food-101. The multiple choice questions generally ask the model to select the taxonomic classification of an image at a given level of the taxonomic tree from a list of four options: the ground truth and 3 similar labels selected a the corresponding level of the taxonomic tree. SigLIP was used to evaluate the similarity between an image and text labels to find potentially difficult options. The authors measured low performance across all the models they tested as measured by Hierarchical Consistent Accuracy and Leaf-Level Accuracy. The models typically returned higher leaf accuracy while returning poor hierarchical results, indicating they made many mistakes along the taxonomic tree. All models did better with ImageNet derived datasets and struggled with the fine-grained data in iNat. GPT-4o generally performed better than open-source models, but still struggled with taxonomy.

**Questions:**

1. How would domain specific models fare in this problem setting, particularly something like BioCLIP that encodes taxonomy by design? Is that the sort of pretraining vs fine tuning referred to at line 286? The [BioCLIP paper](https://openaccess.thecvf.com/content/CVPR2024/html/Stevens_BioCLIP_A_Vision_Foundation_Model_for_the_Tree_of_Life_CVPR_2024_paper.html) reports only that they observe good separation of hierarchical labels in a t-SNE projection. Perhaps this is a good opportunity to probe that.
2. How does the hierarchy depth compare across your datasets? And what refinements were implemented to 'improve label quality and semantic consistency'?
3. Will the code to produce the questions (or the questions themselves) be made available?

**Ethical Concerns:**

["NO or VERY MINOR ethics concerns only"]

**Final Justification:**

The author's provided useful responses to my comments and additional results with BioCLIP. Those new results are quite interesting, adding to the papers already intriguing observations about hierarchical consistency in VLMs.

I reaffirm my original score.

**Limitations:**

The authors put their discussion of limitations in the appendix. I recognize space is limited, but I'd encourage the author to include some of the material (or at least a reference to the section of the appendix) in the main body. The points made in the appendix are worth grappling with.

**Quality:**

3

**Strengths And Weaknesses:**

Strengths:
The authors clearly articulate their experiments and provide a thought provoking probe of visual LLMs understanding of taxonomy. They measure an interesting discrepancy in several large model's hierarchical performance and understanding, even when returning reasonable leaf accuracy. The experiments do a good job of illuminating possible points of breakdown.

Weaknesses:
The authors do not grapple with domain specific models that may provide further insight into the shortcomings they identified.

---

> ### Author Rebuttal · Authors · 2025-07-31
>
> Thank you for your positive feedback on our work. We highly appreciate the detailed and constructive comments. We respond to the identified weaknesses and questions below.
>
> > How would domain specific models fare in this problem setting, particularly something like BioCLIP that encodes taxonomy by design? Is that the sort of pretraining vs fine tuning referred to at line 286? The BioCLIP paper reports only that they observe good separation of hierarchical labels in a t-SNE projection. Perhaps this is a good opportunity to probe that.
>
> We evaluated the performance of the domain-specific models BioCLIP [1] and BioCLIP2 [2] under our experimental setup, and the results will be added to Table 2.
>
> BioCLIP is a foundation model initialized from OpenAI’s CLIP weights and continuously pre-trained on 10 million bio-taxonomy image-text pairs, designed for general organismal biology tasks. Its training data includes the training split of iNat21, and we used the validation split of iNat21 for testing in our setup.
>
> We observed that BioCLIP and BioCLIP2 significantly improve fine-grained recognition performance (Level 6) compared to OpenCLIP and SigLIP, as shown in Table I. This improvement also extends to per-level accuracy, consistent with the clear separation of hierarchical labels reported in the BioCLIP paper using t-SNE visualizations.
>
> **Table I. Level-by-level accuracy and HCA on iNat21-Plant**
>
> | Model     | Level 6 | Level 5 | Level 4 | Level 3 | Level 2 | Level 1 | HCA   |
> |-----------|---------|---------|---------|---------|---------|---------|-------|
> | SigCLIP   | 18.84   | 35.08   | 27.19   | 18.12   | 46.61   | 62.18   | 0.463 |
> | OpenCLIP  | 28.11   | 41.99   | 34.85   | 16.04   | 16.58   | 68.81   | 0.192 |
> | BioCLIP   | 89.47   | 79.53   | 61.62   | 44.73   | 73.52   | 69.94   | 11.67 |
> | BioCLIP2   | 95.26   |  91.48  |82.68  | 74.26   |   65.59 | 96.75  | 37.91 |
>
> However, the gains in hierarchical consistency of BioCLIP are modest on iNat21-Plant, as the accuracy at intermediate hierarchy levels remains relatively low, which results in incorrect hierarchy paths (low HCA). Moreover, although BioCLIP leverages taxonomic labels during training, its primary objective is to improve leaf-level (species) classification. While some hierarchical labels (e.g., Animalia, Chordata, Aves, Passeriformes, Corvidae, Pica hudsonia) are included in the training data, they account for only a small portion of the overall dataset. As a result, the model encodes some degree of hierarchical structure through pretraining. Still, it is not explicitly optimized for enforcing hierarchical consistency, which is the primary focus of our evaluation.
>
> We also conducted experiments on a simpler dataset, CUB-200, which contains only four hierarchical levels, as shown in Table II. On this dataset, BioCLIP achieves substantial improvements. Since the accuracy at each level is high, the HCA also increases significantly. However, the persistent gap between HCA and fine-grained accuracy suggests limited hierarchical visual consistency.
>
> **Table II. Level-by-level accuracy and HCA on CUB-200**
>
> | Model     | Level 4 | Level 3 | Level 2 | Level 1 | HCA   |
> |-----------|---------|---------|---------|---------|-------|
> | SigCLIP   | 73.84   | 51.61   | 58.80   | 82.01   | 23.18 |
> | OpenCLIP  | 80.39   | 30.67   | 28.31   | 35.55   | 4.31  |
> | BioCLIP   | 82.83   | 90.65   | 78.75   | 83.76   | 51.48 |
> | BioCLIP2   |  92.94  |  93.63  |  79.67 |  78.86 |  55.80|
>
> [1]  Stevens, Samuel, et al. "Bioclip: A vision foundation model for the tree of life." Proceedings of the IEEE/CVF conference on computer vision and pattern recognition. 2024.
>
> [2] Gu, Jianyang, et al. "Bioclip 2: Emergent properties from scaling hierarchical contrastive learning." arXiv preprint arXiv:2505.23883 (2025).
>
> > How does the hierarchy depth compare across your datasets?
>
> We provided an overview of the six taxonomies and four datasets used to construct our VQA tasks in Table 1 in the paper, reproduced below. The table summarizes the number of hierarchy levels, leaf nodes, test images, and the distribution of categories across levels. This illustrates the variation in hierarchy depth across datasets, ranging from 4 to 11 levels.
>
> **Table III. Overview of the datasets and their hierarchical statistics**
>
> | Dataset               | # Levels | # Leaf Nodes | # Test Images | Hierarchy Distribution                |
> |------------------------|----------|--------------|----------------|----------------------------------------|
> | CUB-200-2011      | 4        | 200          | 5,794          | 13-37-124-200                          |
> | iNaturalist-Plant |  6      | 4,271        | 42.71K         | 5-14-85-286-1700-4271              |
> | iNaturalist-Animal| 6     | 5,388     | 53.88K         | 6-27-152-715-2988-5388               |
> | ImageNet-Animal     | 11      | 397   | 19.85K    | 2-10-37-81-123-81-65-41-64-34-2    |
> | ImageNet-Artifact | 7        | 491          | 24.55K       | 5-40-147-204-162-62-44            |
> | Food-101          | 4        | 84           | 21.00K          | 6-29-40-24                                     |
>
>
> > And what refinements were implemented to 'improve label quality and semantic consistency'?
>
> To refine label quality and ensure semantic consistency, we relied on GPT-4o and Wikipedia to carefully examine the hierarchical relations in each taxonomy. There are several types of refinement we made:
>
> * Re-verify Hierarchical Relationships: For example, in the original WordNet hierarchy, "indigo bunting" is misclassified under "finch". However, based on established taxonomy, it belongs to the cardinal family. We corrected its hierarchical path to:  animal → vertebrate → bird → oscine → cardinal → indigo bunting.  We detected these errors using GPT-4o and then validated them using reliable taxonomies in iNaturalist and Wikipedia.
>
> * Remove Ambiguous Path: For example, in WordNet, "tusker" is assigned to the overly coarse path:  animal → vertebrate → mammal → tusker.  However, "tusker" is merely a colloquial term for an elephant, and WordNet already includes a more fine-grained and taxonomically accurate path for "elephant":  animal → vertebrate → mammal → placental → elephant → African elephant. We removed the "tusker" category, as it lacks specificity and overlaps with the existing, more precise elephant hierarchy.
>
> * Correct Subordinate Relationships Within the Same Hierarchy Level: In the ImageNet-Artifact dataset, the hierarchy provided by WordNet exhibits notable semantic inconsistencies, particularly where concepts at the same hierarchical level implicitly reflect subordinate relationships. For instance, under the category "device," concepts such as "machine," "instrument," "musical instrument," and "mechanism" are listed as siblings. However, "musical instrument" is a subtype of "instrument," making the latter a hypernym of the former. Treating these as peers can lead to ambiguous or conflicting answers when classifying an image, as both may be considered valid even though one is semantically nested within the other. To resolve these issues, we used GPT to analyze whether sibling category pairs exhibited valid hypernym-hyponym relationships systematically. We refined or removed problematic intermediate nodes and eliminated leaf nodes associated with overly coarse or semantically inconsistent hierarchy paths.
>
> These refinements enhance the overall quality and coherence of the taxonomy, supporting more accurate and interpretable hierarchical reasoning, and we will release the refined taxonomies together with the code and VQA tasks publicly.
>
> > Will the code to produce the questions (or the questions themselves) be made available?
>
> Yes, we will release the code and questions after the double-blind review process.
>
> > Encourage the author to include some of the material (or at least a reference to the section of the appendix) in the main body. The points made in the appendix are worth grappling with.
>
> Thank you for the suggestion. We will incorporate additional information into the main body in the revised version and include a specific reference to the Limitations section in the appendix.

---

### Official Review · Reviewer_C5mQ · 2025-07-02

**Clarity:** 3
**Significance:** 2
**Originality:** 3
**Rating:** 3
**Confidence:** 4

**Summary:**

The paper analyzes off-the-shelf VLM performance on hierarchical visual understanding. By hierarchical, the authors actually imply taxonomical classification of animals and plants, which is a subtle but significant difference in the claim of the paper. The experiments conducted are meaningful and sound, with multiple datasets and configurations tested. The authors are convinced that LLMs are mostly responsible for misclassifications rather than the VLMs or more specificaly the vision tokens are ample for this task.

**Questions:**

See weakness.

**Ethical Concerns:**

["NO or VERY MINOR ethics concerns only"]

**Final Justification:**

I have issues with the 'hierarchy' definition which was not clarified well in the paper.
No deeper insight into VLMs except measuring how well they follow world knowledge.
Presentation issues.

**Limitations:**

Yes.

**Paper Formatting Concerns:**

None.

**Quality:**

3

**Strengths And Weaknesses:**

Strengths:
1. Large-scale, hierarchy-aware benchmark: curates about 1 M  data points across six taxonomies (~10k leaf classes) and evaluates with both Acc_leaf and HCA, giving the community the first systematic testbed for fine-grained hierarchical VQA
2. Clean ablation pinpoints the bottleneck: independent linear-probe study shows vision tokens are already linearly separable at every taxonomic level, shifting blame from the image encoder to the language head and sharpening the paper’s core insight

Weakness:
1. My primary concern is with the clarification of the objective of the work. Even though the authors claim that VLMs are bad at "Hierarchical Visual Understanding", a more accurate version would be that VLMs are bad at Taxonomic understanding in animal images. The 1M VQA pair datasets are heavily dominated by animal images and taxonomic-related questions, with only Imagenet-Artifact being a general-purpose dataset.
2. While I understand the purpose of probing the vision tokens - I have siginificant reservations around experiment in Section 3.2. Even though linear probes classify each level correctly, the three classifiers are trained separately for kingdom, phylum, etc. so they never have to respect parent-child constraints. A probe can be 100 % accurate at every level yet still assign mutually incompatible ancestors (e.g. Invertebrate parent of Fish) because cross-level consistency is never evaluated. On similar lines, a better experiment would be to probe vision tokens in a path like (Fish|Inertebrate) and (Fish|Vertebrate). If the vision part is still solid in differentiating them, that would conclude that the vision tokens are not responsible.
3. Table-4: "Our finetuning data consists of VQA tasks constructed from iNat21-Plant’s 291 training set, covering 3,771 species nodes in the taxonomy instead of the full 4,271 species nodes". Well this observation is pretty much in-line wuth what should be expected? If the VLMs are not able to understand the difference b/w Annelids and Arthropods, how can they be "taught" this by fine-tuning on *plant species*? As you see in Table-4, performance on iNat21-Plant improves by a lot as compared to CUB. I am sure more training data and prompt structuring can push gains even further but never improve animal specie performance.
4. (minor) Instead of just measuring raw accuracy, can the authors also report the accuracies measured through VQAScore? There is usually a 5-6% jump when actually looking at probabilities than tokens.
5. Lastly it would be interesting to see what level the LLM/VLM misses on the most. How different is the probability of token prediction of vertebrate/invertebrate?

---

> ### Author Rebuttal · Authors · 2025-07-30
>
> We greatly appreciate your detailed feedback and insightful suggestions. We sincerely hope that our responses below will effectively address your concerns.
>
> >The 1M VQA pair datasets are heavily dominated by animal images and taxonomic-related questions, with only Imagenet-Artifact being a general-purpose dataset.
>
> We would like to clarify that our work goes beyond taxonomic understanding in animal images. Specifically, our evaluation includes:
>
> * ImageNet-Artifact, which covers a wide variety of man-made objects and serves as a general-purpose hierarchy benchmark.
>
> * iNaturalist-Plant, which focuses on plant taxonomy.
>
> * Food-101, which introduces a hierarchy of food categories. (Please refer to Table 11 in the supplementary material for more evaluation results.)
>
> The detailed data distribution of the evaluated datasets is illustrated in Table 1 in the main body. For additional clarity, we provide the complete statistics on #VQA pairs from each dataset below:
>
> | Dataset               | # VQA Pairs |
> |------------------------|-------------|
> | CUB-200-2011      |    23k         |
> | iNaturalist-Plant |     256k        |
> | iNaturalist-Animal|      323k       |
> | ImageNet-Animal |     159k        |
> | ImageNet-Artifact|       123k      |
> | Food-101 |     84k        |
> | Total |     968k        |
>
> As shown in the table, animal-related and non-animal-related VQA pairs are balanced at approximately a 1:1 ratio, ensuring that our evaluation is not biased toward animal taxonomies.
>
> Finally, we studied the datasets separately in the paper rather than combining them into an overall corpus, further avoiding the over-dominance of any particular dataset.
>
> > A probe can be 100 % accurate at every level yet still assign mutually incompatible ancestors (e.g. Invertebrate parent of Fish) because cross-level consistency is never evaluated. On similar lines, a better experiment would be to probe vision tokens in a path like (Fish|Inertebrate) and (Fish|Vertebrate). If the vision part is still solid in differentiating them, that would conclude that the vision tokens are not responsible.
>
> We have actually evaluated the linear probes' cross-level consistency in Figure 3. We appreciate the reviewer’s thoughtful comment and fully agree that evaluating the cross-level consistency is essential when probing visual tokens.
>
> While we train separate linear classifiers for each taxonomy level, our probing analysis does not focus on per-level accuracy. Instead, as shown in Figure 3 of Section 3.2, we report results using Hierarchical Consistency Accuracy (HCA). This metric explicitly measures whether the predicted labels across all levels form a correct hierarchical path.
>
> Figure 3 shows that vision tokens consistently achieve high HCA scores, even though the classifiers of different taxonomy levels are independently trained. This suggests that visual embeddings preserve discriminative and structurally aligned features with respect to the underlying hierarchy.
>
> These findings support our conclusion that visual representations are not the bottleneck in hierarchical understanding. We will revise Section 3.2 to emphasize our use of HCA and its implications more clearly.
>
> > Table-4: "Our finetuning data consists of VQA tasks constructed from iNat21-Plant’s 291 training set, covering 3,771 species nodes in the taxonomy instead of the full 4,271 species nodes". Well this observation is pretty much in-line with what should be expected?
>
> Firstly, we would like to highlight that fine-tuning on iNat21-Plant data also improves performance on animal datasets, which is a non-trivial and unexpected finding.
>
> As shown in Table 4:
>
> * ImageNet-Animal HCA increases from 56.00 to 58.62
>
> * iNat21-Animal HCA increases from 19.43 to 23.38
>
> * CUB-200 HCA increases from 43.76 to 46.17
>
> This improvement is observed not only in HCA but also in leaf-node accuracy. The gains extend beyond the plant domain, suggesting that the fine-tuning leads to more generalizable hierarchical consistency rather than just domain-specific memorization. We acknowledge that this improvement is surprising, and we plan to investigate the underlying causes in future work. One possible hypothesis is that fine-tuning enhances the LLM’s general ability to reason about hierarchical structures, hence benefiting hierarchical understanding tasks even in unseen domains.
>
> Additionally, we would like to emphasize that the goal of our fine-tuning experiments is not merely to boost performance on the dataset used for fine-tuning. The non-trivial and more interesting finding is that the LLM’s (text) HCA (Table 5) increases more than the (visual) HCA (Table 4) after fine-tuning. This reinforces that LLMs are the primary bottleneck in VLLMs' hierarchical visual understanding, and one ultimately has to improve LLMs’ (text) taxonomy knowledge to boost VLLMs’ (visual) hierarchical consistency.
>
> > (minor) Can the authors also report the accuracies measured through VQAScore?
>
> We follow the setup in VQAScore [1] to formulate our experiment as a task of matching an image to four candidate captions, aligning with our original design of a four-choice QA. Each caption takes the form: "Does this figure show {level label}? Please answer yes or no." For each image, we conduct four runs: one with the ground-truth label and three with similar but incorrect labels. We then compute the VQAScore for each run and check whether the caption with the highest score corresponds to the ground-truth label. If so, the prediction is considered correct; otherwise, it is counted as incorrect.
>
> We report both level-wise accuracy and HCA for our metric (Raw Accuracy) and VQAScore on the iNat21-Plant dataset using Qwen2.5-VL-7B in the table below. For level-wise accuracy, Raw Accuracy and VQAScore have similar results. VQAScore achieves slightly higher accuracy at levels 1 and 2, but lower accuracy from levels 3 to 6. In terms of HCA, no improvement is observed when using VQAScore compared to Raw Accuracy.
>
>
> | Metric    | Level 6 | Level 5 | Level 4 | Level 3 | Level 2 | Level 1 | HCA   |
> |-----------|---------|---------|---------|---------|---------|---------|-------|
> | Raw Accuracy |    41.61  |   61.04 |   69.35 |   69.95 |   90.92 |   98.81 | 17.76 |
> | VQAScore  | 40.45   | 55.94   | 62.84   | 63.00   | 91.11   | 99.21   | 13.19 |
>
> Additionally, we have conducted a similar experiment in Section C.1.4 of the supplementary material, using four statements that require binary yes-or-no responses. We will add the above experiments to that section in the revised paper.
>
> [1] Lin, Zhiqiu, et al. "Evaluating text-to-visual generation with image-to-text generation." European Conference on Computer Vision. Cham: Springer Nature Switzerland, 2024.
>
>
> > Lastly it would be interesting to see what level the LLM/VLM misses on the most.
>
>
> Thank you for this insightful suggestion. We report the level-wise accuracy in the table below, Figure 4 in the paper shows the text HCAs of LLMs at different taxonomic levels, and Table 18 (Row 3 and 4 below) in the supplementary material reports the level-wise HCA of VLLMs and their LLMs, all on iNat21-Plant. For the VLLM, the lowest performance occurs at the leaf level (Level 6, species), while for the LLM, the weakest performance is observed at Level 3 (order).
>
>
> Further, for the visual HCA (Row 3), we observe a significant drop in performance from Level 2 to Level 3, with a decrease of approximately 25%, suggesting that Level 3 (order) is where the VLLM struggles the most hierarchically, aligning with LLM’s low performance at Level 3.
>
>
> Finally, for the text HCA (Row 4), the most substantial performance drop occurs from Level 2 to Level 3, suggesting that the LLM struggles to reason upward through the hierarchy, particularly from the “order” level to higher levels such as “class” and “phylum”.
>
> | Metric            | Level 6 | Level 5 | Level 4 | Level 3 | Level 2 | Level 1 | Mean |
> |------------------|--------:|--------:|--------:|--------:|--------:|--------:|--------:|
> | Visual  Level-wisel Accuracy |    41.61  |   61.04 |   69.35 |   69.95 |   90.92 |   98.81 | 71.95 |
> | Text  Level-wise Accuracy  |    N/A   |   99.79 |   87.18 |   78.58 |   88.13 | 94.49 |91.36 |
> | Visual HCA    |   17.67 |   35.15 |   51.86 |   65.32 |   90.53 |   98.81 | N/A |
> | Text HCA   |   64.22 |   60.06 |   79.78 |   65.99 |   99.86 | N/A | N/A|
> | ProbDiff  |    0.2485  |   0.1904 |   0.1613 |   0.1497 |   0.0489 |   0.0053 | 0.1340 |
> | Text ProbDiff  |    N/A  |   0.0012 |   0.0430 |   0.0845 |   0.0229 |   0.0143 | 0.0332 |
>
>
>
> > How different is the probability of token prediction of vertebrate/invertebrate?
>
> We conducted experiments on iNat-21 using Qwen2.5-VL-7B to calculate the level-wise probability difference (ProbDiff) between the model's predictions and the ground truth, using both visual and text-only QA pairs. The results are shown in the table above.
>
> The VLLM’s ProbDiff is larger at each level than that of the LLM in general; specifically, it reaches 0.2458 at the leaf level,  aligning with the poor visual level-wise accuracy at level 6. The LLM’s small ProbDiff probably explains why LLMs benefited from our finetuning more than VLLMs as it is intuitively easier to correct the small ProbDiff than VLLM’s big ones.

---

> > ### Comment · Reviewer_C5mQ · 2025-08-04
> > **Thank you for the experiments**
> >
> > I appreciate all the new experiments.
> > I am still unconvinced by the "hierarchical" nature of the problem, which to me seems like 'hierarchy' just corresponds to semantic hierarchy rather than visual hierarchy, which does not really shed any insight into VLM's architecture but only to their world knowledge.
> >
> > However, I see some merit in the work wherein VLMs are evaluated to determine whether they respect the semantic hierarchy. The paper can also benefit from improved presentation with a clear definition of what 'hierarchy' is.
> >
> > I would like to maintain my ratings.

---

> > > ### Author Response · Authors · 2025-08-04
> > > **Follow up with a clarification question**
> > >
> > > Could you elaborate on the visual hierarchy? What does this concept refer to? Thank you so much.!

---

> > > > ### Author Response · Authors · 2025-08-06
> > > >
> > > > Gentle reminder: Since the concept of "visual hierarchy" was not in the original review and instead newly composed in the discussion, could the review help us understand it before we respond to it? Much appreciated!

---

> > > > > ### Comment · Reviewer_C5mQ · 2025-08-08
> > > > > **Follow up**
> > > > >
> > > > > By visual hierarchy, I imply how *visual structures* are learned in VLMs i.e. how does an "animal" image relates to a "cat" image. The paper, in its current state, discusses the labels and names of the classes and the types of images, which is more about whether the latent representations map to the correct labels and names. In my opinion, this is not visual understanding - merely classification (where class labels are ordered in a structured way).
> > > > >
> > > > > Nevertheless, I believe the paper can go through another round of major revision with a clear contribution to taxonomic classification and not a broad "Hierarchical Visual Understanding".
> > > > >
> > > > > I maintain my rating.

---

> ### Author Response · Authors · 2025-08-08
> **"Visual structure" & "mapping to labels" are equally significant research questions**
>
> Thank you, Reviewer C5mQ, for the response!
>
> > "Visual structure (e.g., how "animal" relates to "cat")" **vs.**  "mapping to labels"
>
> Both are significant research questions in our humble opinion, and we sincerely request the reviewer to *not* undermine the latter (mapping) and please assess our work in the context of the latter. If we follow the reviewer's suggestion to make major revisions to the paper, we'd have to tackle two big questions in one work --- probably not a good idea as it will blur the focus and main message. Moreover, it might be too early to study the former (structure) given that our work showed that existing VLMs, including GPT-4o, are still bad on the latter --- their "mapping" does not observe the hierarchical consistency.
>
> RE Terminology: We used "visual understanding" to refer to the latter (mapping) rather the former (structure) following the literature [1,2]. To avoid ambiguity, probably the future work focusing on the former can use the language suggested by the reviewer, "visual structure", or something related to VLM interpretability.
>
> Looking forward to hearing from the reviewer soon.
>
> [1] Guo Y, Liu Y, Oerlemans A, Lao S, Wu S, Lew MS. Deep learning for visual understanding: A review. Neurocomputing. 2016 Apr 26;187:27-48.
>
> [2] Sugiyama K, Tagawa S, Toda M. Methods for visual understanding of hierarchical system structures. IEEE Transactions on Systems, Man, and Cybernetics. 2007 Nov 12;11(2):109-25.

---

### Note · Authors · 2025-08-12

Dear Reviewers, ACs, SAC, and PCs,

We learned a lot from your suggestions of BioCLIPs, clarification of opensource vs. proprietary LLMs, VQAScores, and hyperbolic image-text representations, and we will improve the work accordingly.

We are glad to see that our initial rebuttal was responsive and satisfying to most of your questions. The follow-up discussions were mostly about lingering smaller points reiterated below, and we hope those will not diverge your assessment from the paper’s main message.
* Reviewer C5mQ: We recognize that both visual structure and mapping to semantics are important research questions. We respectfully suggest the reviewer evaluate our work in the context of mapping, which is the primary focus of this paper. Addressing both questions within a single work could dilute the main points.
* Reviewer 71bs: We would like to re-emphasize that the goal of the linear probing is not merely to compare their performance with zero-shot probing, but rather to pinpoint the cause of the poor visual hierarchical consistency. Providing classifiers with label space information does not guarantee strong probing performance unless the visual embeddings are well-structured. Additionally, the linear probing rely on per-level taxonomy labels as one-hot vectors without hierarchical constraints, so its high hierarchical consistency primarily reflects the quality of the visual embeddings.

Thank you for your time and efforts,
Authors

---

### Decision · Program_Chairs · 2025-09-17

**Decision:**

Reject

**Comment:**

This submission received highly diverse scores (two positive ones and two negative ones). The positive side thinks that asks an important question (hierarchical visual knowledge in VLLMs), and comes up with an interesting and important finding. The experiments are comprehensive and convincing. The majors concerns (after rebuttal ) from the negative side remains. One primary concern is with the clarification of the objective of the work  and unconvinced "hierarchical" nature of the problem. The others are about the insufficient justification of the proposed approach, which is discussed to the very end of the rebuttal period.

After carefully reading the paper, the rebuttal and the discussions,  I agree with the reviewers who assigned the negative scores, as the question is not well defined and justified. As the authors investigated an important and interesting question, I suggest them to carefully consider the concerns raised by reviewers and submit their work to the next conference.